

# An intercomparison of EarthCARE cloud, aerosol and precipitation retrieval products

Shannon L. Mason[1,2], Jason N. S. Cole[7], Nicole Docter[9], David P. Donovan[4], Robin J. Hogan[1,3],
Anja Hünerbein[8], Pavlos Kollias[5,6], Bernat Puigdomènech Treserras[6], Zhipeng Qu[7], Ulla Wandinger[8],
and Gerd-Jan van Zadelhoff[4]

[1]European Centre for Medium-range Weather Forecasts (ECMWF), Reading, UK
[2]National Centre for Earth Observation, University of Reading, Reading, UK
[3]Department of Meteorology, University of Reading, Reading, UK
[4]Royal Netherlands Meteorological Institute (KNMI), De Bilt, Netherlands
[5]Stonybrook University, Stonybrook, New York, USA
[6]McGill University, Montreal, Quebec, Canada
[7]Environment and Climate Change Canada, Toronto, Ontario, Canada
[8]Leibniz Institute for Tropospheric Research (TROPOS), Leipzig, Germany
[9]Free University of Berlin, Berlin, Germany

**Correspondence:** Shannon Mason (shannon.mason@ecmwf.int)

**Abstract.** The mission of the Earth cloud, aerosol and radiation explorer (EarthCARE) mission to observe cloud, aerosol, precipitation and radiation using four complementary instruments requires the development of many single-instrument and synergistic algorithms for the retrieval of geophysical quantities. The retrieval products employ one or more of the cloud profiling radar (CPR), atmospheric lidar (ATLID) and multispectral imager (MSI), while the broadband radiometer (BBR)

places the retrieved quantities in the context of the atmospheric radiation budget. To facilitate the development and evaluation of the ESA EarthCARE production model prior to launch, sophisticated instrument simulators have been developed to produce realistic synthetic EarthCARE measurements from the output of cloud-resolving model simulations. While acknowledging that the physical and radiative representation of cloud, aerosol and precipitation in the test scenes are based on numerical models, the opportunity to perform a detailed evaluation wherein the model "truth" is known has provided rare insights into

the performance of EarthCARE's instruments and retrieval algorithms. This level of omniscience will not be available for the evaluation of in-flight EarthCARE retrieval products, even during validation activities coordinated with ground-based and airborne measurements. In this study we intercompare EarthCARE retrieval products from within the ESA production model both statistically across all simulated EarthCARE granules, and using timeseries of data from an individual scene. The comparison between the retrieved quantities helps to illustrate the strengths and limitations of the single-instrument retrievals,

and the degrees to which the synergistic retrieval and composite products can represent the entire atmosphere of clouds, aerosols and precipitation.

We show that radar-lidar synergy has the greatest impact in ice clouds; when compared with single-instrument radar and lidar retrievals, the synergistic ATLID-CPR-MSI cloud, aerosols, and precipitation (ACM-CAP) product accurately retrieves profiles of both ice water content and effective radius. While liquid cloud is difficult to detect directly from spaceborne remote sensors,



especially in complex and layered scenes, the synergistic retrieval benefits from combined constraints from lidar backscatter, solar radiances and radar path-integrated attenuation, but still exhibits a high degree of random error. For precipitation retrievals, the CPR cloud and precipitation product (C-CLD) and ACM-CAP have similar performance when well-constrained by CPR measurements. The greatest differences are in coverage, with ACM-CAP reporting retrievals in the melting layer, and in heavy precipitation where the radar is dominated by multiple scattering and attenuation). Aerosol retrievals from ATLID compensate for a high degree of measurement noise in a number of ways, with the ATLID extinction, backscatter and depolarization (A-EBD) product and ACM-CAP demonstrating similar performance in the test scenes. The multispectral imager (MSI) cloud optical properties (M-COP) product performs very well in unambiguous cloud layers; similarly, the MSI aerosol optical thickness (M-AOT) product performs well where the possibility of contamination by cloud signal is very low. A summary of the performance of all retrieval products is provided, and may help to inform the selection of EarthCARE data products by future users.

## 1 Introduction

With its mission to measure clouds, aerosols and precipitation and their radiative effects using two active and two passive instruments aboard a single platform, EarthCARE is the most complex of ESA's Earth Explorer satellites to date. EarthCARE's four instruments are the 355nm Atmospheric Lidar (ATLID), 94-GHz Cloud Profiling Radar (CPR), the Multi-Spectral Imager (MSI) and the Broadband Radiometer (BBR), each fully described in Wehr et al. (2023). The ESA EarthCARE production model (Eisinger et al., 2023) includes 14 single-instrument (L2a) and 11 synergistic (L2b) data products, which will present the corrected measurements from all instruments, interpret those measurements to describe the spatial distributions and classifications of hydrometeors and aerosols through the atmosphere, retrieve the bulk quantities and microphysical properties of clouds, precipitation and aerosols, and finally place each scene in its broadband radiative context. As part of the development of EarthCARE's L2 processors, an unprecedented effort has been directed to the simulation of EarthCARE measurements based on high-resolution numerical weather forecasts (Donovan et al., 2023a; Qu et al., 2022). These simulated test scenes have provided the basis for developing and testing the L2 processors ahead of EarthCARE's launch, and facilitate an evaluation of retrieved geophysical quantities wherein the "true" physical quantities are known.

Each L2 processor and its data products have been described and evaluated using the simulated test scenes in dedicated publications. In this study we present an intercomparison of geophysical retrievals describing the bulk amount and microphysical properties of hydrometeors and aerosols:

- ATLID retrievals of profiles of ice clouds (A-ICE) and aerosols (A-AER and A-EBD) are described in Donovan et al. (2023b), and the aerosol layer descriptor (A-ALD) in Wandinger et al. (2023);

- CPR retrievals of ice and liquid clouds, snow and rain (C-CLD) are described in Mroz et al. (2023);

- Passive retrievals of cloud (M-CLD) are described in Hünerbein et al. (2023b, a) and of aerosol (M-AOT) in Docter et al. (2023);



– Retrievals from EarthCARE's active instruments—ice clouds from A-ICE and C-CLD, snow, liquid clouds and rain from C-CLD, and aerosols from A-EBD—are composited into a unified cloud, aerosols and precipitation product (ACM-COM; Cole et al., 2022); and,

– The corrected measurements from active and passive sensors provide the synergistic inputs for a simultaneous and unified retrieval of all cloud, aerosol and precipitation (ACM-CAP; Mason et al., 2023).

The production of both single-instrument (L2a) and synergistic (L2b) retrievals builds some redundancy into the production model, and provides the capacity for an immediate internal verification of EarthCARE products based on intercomparison between geophysical retrieval products within the same processing chain.

Prior to launch the evaluation of EarthCARE retrieval products has been limited to simulated EarthCARE scenes and measurements from previous satellites or field campaigns that can be used as proxies for EarthCARE. After launch, validation efforts will depend on correlative data products from dedicated field campaigns and supersites; however, with multiple single-instrument and synergistic products available, one valuable form of verification that can be carried out using both simulated scenes and in-flight data is by detailed intercomparison of EarthCARE retrievals. Intercomparisons have been a key part of

the algorithm development workflow, and inform planning for verification activities during the Commissioning Phase. The description of the EarthCARE production model (Eisinger et al., 2023) and the papers describing the individual processors will provide important complementary information to this study, which is intended as a reference by which users of EarthCARE data can understand the differences between products, their performance as currently understood when applied to the simulated test scenes, and to select data products suited to their purposes. We limit the intercomparison to retrieved geophysical quanti-

ties and properties of clouds, precipitation and aerosols: an intercomparison of single-instrument and synergistic detection and target classification products is made in Irbah et al. (2023), while radiative fluxes and heating rates derived from two sets of retrieval products are compared in Barker et al. (2023).

In Section 2 we outline the EarthCARE retrieval products that are included in this study, and describe the quantities from the numerical weather model against which they are evaluated. The evaluation (Section 3) is separated into retrievals of ice cloud

and snow (Section 3.1), liquid clouds (Section 3.2), rain (Section 3.3) and aerosols (Section 3.4). A summary of the respective performance of single-instrument and synergistic retrievals products, and some discussion and concluding remarks, are made in Section 4.

## 2  Data products

Two sets of data products are used to carry out the present evaluation and intercomparison. The "model truth" data products

(Section 2.1) have been postprocessed from the numerical model data fields used as inputs for the simulated test scenes. These are non-standard data products that are specific to this activity. The retrieval products (Section 2.2) are the output of the official ESA EarthCARE L2 processors at the time of writing, which will closely resemble the data structure and variable naming conventions of the L2 products available after EarthCARE launch.



## 2.1 Model truth

The creation of the three EarthCARE test scenes based on Environment and Climate Change Canada (ECCC)'s high-resolution Global Environmental Multiscale (GEM) model and Copernicus Atmospheric Modelling Service (CAMS) aerosols forecasts is described in Qu et al. (2022), and the simulation of EarthCARE measurements is described in Donovan et al. (2023a). The thermodynamical, cloud and precipitation fields are based on the GEM model, and are merged with the CAMS aerosol fields. The merged and modified quantities describing the properties of the atmosphere, clouds, precipitation and aerosols is referred

to in this paper as the "model truth". The model truth has been collated in two data products, which are archived alongside the L2 products (see Data Availability).

While evaluating the L2 products against the model truth in these simulated scenes provides many insights into the performance of the retrievals, we stress that the numerical models and instrument simulators rely on approximations to the same quantities and properties as the retrievals (e.g. ice and rain fall speeds, ice density, hydrometeor size distributions and number

concentrations, etc.). Apparent errors or biases in the retrievals presented in this paper may therefore be due to differences in assumptions underlying the model truth, and should not be interpreted as unambiguous deficits in the retrieval algorithms. These uncertainties can only ultimately be quantified with in-flight EarthCARE observations and correlative measurements from validation activities.

## 2.2 Retrievals

The EarthCARE L2 production model is described in detail in Eisinger et al. (2023). In this section we provide brief descriptions of selected products which include geophysical retrievals of cloud, aerosol and precipitation.

Table 1 shows the L2 data products from which the retrieved quantities can be intercompared. Not all of these quantities are shown in this paper: single-instrument and synergistic profiling macrophysical products (A-TC, C-TC and AC-TC) intercompared in Irbah et al. (2023), while the radiative quantities are evaluated as part of the radiative closure assessment study

(Barker et al., 2023). Among the geophysical retrieval products, we have not included the synergistic aerosol column product AM-ACD (Haarig et al., 2023) in this intercomparison.

### 2.2.1 MSI retrievals

The passive multi-spectral imager (MSI) measures cloud, aerosols and the surface with four solar channels from the visible (670 nm) to the shortwave infrared (2.2 $\mu$m), and three thermal-infrared channels (8.5 $\mu$m to 12.5 $\mu$m), with 500-m spatial

resolution across a 150-km swath (Wehr et al., 2023).

The L2a MSI Cloud Optical and Physical Properties (M-COP; Hünerbein et al., 2023a) product reports cloud optical thickness, effective radius and derived cloud water path estimates and their uncertainties based on MSI's visible, shortwave infrared and thermal-infrared channels. The thermodynamic phase, structure and degree of certainty of cloudy pixels are differentiated using "cloud phase" and "cloud mask" variables which are part of the L2a MSI Cloud mask product (M-CM; Hünerbein et al.,



2023b). Unless otherwise stated, in this paper we interpret cloud water path pixels as liquid water path when the cloud phase is "liquid" or "supercooled/mixed", and as ice water path when the cloud phase is "ice" or "overlap".

The L2a MSI Aerosol Optical Thickness (M-AOT; Docter et al., 2023) product includes aerosol optical thickness over ocean and land at 670nm, and over ocean at 865nm, based on MSI's visible, near infrared and shortwave infrared channels. While both M-COP and M-AOT are provided across the MSI swath, for this study we present MSI retrievals at the nadir pixel only

for intercomparison with the active instruments.

### 2.2.2 ATLID retrievals

The 355-nm atmospheric lidar (ATLID) (Wehr et al., 2023) with high-spectral resolution lidar (HSRL) capability, detects clouds and aerosols. All L2 products downstream of the ATLID feature mask (A-FM; van Zadelhoff et al., 2023b) are postprocessed to a grid with 100-m vertical and 1-km along-track resolution. This grid defines the nadir curtain of the joint standard grid (JSG),

onto which the data are mapped for all L2a ATLID and MSI products, and L2b synergistic data products such as ACM-CAP and ACM-COM.

The L2a ATLID aerosol product (A-AER; Donovan et al., 2023b) estimates the profile of extinction from targets identified as weak scatterers by A-FM (i.e. aerosols and optically thin ice clouds), using a long averaging window (150 km in the present configuration) to increase the signal-to-noise ratio. Due to its long averaging window, the aerosol retrievals in A-AER have a

characteristically smooth distribution. The L2a ATLID extinction, backscatter and depolarization product (A-EBD; Donovan et al., 2023b) is based on an optimal estimation retrieval of extinction profiles, constrained by ATLID's Rayleigh (i.e. molecular) and Mie (i.e. particulate) backscatter channels. The smoother A-AER provides a "first pass" aerosol retrieval as an input to A-EBD, which retrieves smaller-scale features where justified by stronger particulate backscatter signals.

The A-EBD product provides an input to downstream ATLID products that further interpret the extinction profiles to estimate

the physical properties of aerosols and ice. The L2a ATLID aerosol layer descriptor (A-ALD; Wandinger et al., 2023) identifies aerosol layers and their layer-averaged properties. A-ALD provides a total column aerosol optical thickness based on A-EBD, for comparison to the total optical thickness of identified aerosol layers to quantify how much a layer-based approach underestimates the contributions of unresolved aerosols in the profile. The L2a ATLID retrieval of ice cloud (A-ICE; Donovan et al., 2023b) combines an estimate of extinction from A-EBD with the temperature-dependent relation of ice water content and

hence effective radius. The extinction coefficient is provided in liquid clouds, the properties of liquid clouds are not retrieved from ATLID alone, due to the rapid extinction of the lidar in liquid layers.

All ATLID aerosol products include at least one aerosol classification variable, in which six tropospheric aerosol mixtures (i.e. dust, sea salt, continental pollution, smoke, dusty smoke, dusty mix) are distinguished; however, we note that these classifications are not identical between products, and the most robustly processed aerosol classification is included in the A-TC

product. In general, this involves assigning a higher proportion of volumes to one of the "unknown" classes when the signal is difficult to interpret: these volumes are often at the edges of cloud features or where ATLID is nearly completely attenuated. For consistency, we therefore select for aerosols across the A-AER and A-EBD products using the same A-TC classification





variable, and describe where this has a significant impact on the evaluation. We recommend that users use the most advanced target classification variable available: A-TC or even the synergistic product AC-TC.

### 2.2.3 CPR retrievals


The W-band (94 GHz) cloud profiling radar (CPR) has Doppler capability and approximately 5 dBZ more sensitivity than *CloudSat* (Wehr et al., 2023), and provides measurements of clouds and precipitation. CPR measurements are postprocessed onto a 100-m vertical grid (oversampled from 500-m vertical pulse length) with 1-km along-track resolution. While the CPR L2a products are provided on their own spatial grid, for the L2b synergistic products (e.g. ACM-CAP and ACM-COM) CPR data are mapped to the Joint Standard Grid using a nearest-neighbour interpolation.


The L2a CPR cloud and precipitation product (C-CLD; Mroz et al., 2023) employs an optimal estimation retrieval algorithm to retrieve ice cloud, snow, rain and liquid clouds from the radar reflectivity, mean Doppler velocity, and path-integrated attenuation measurements of the CPR. Due to its high sensitivity, CPR is expected to detect some amount of non-precipitating liquid cloud, but the radar measurements are dominated by larger precipitating particles when present. C-CLD includes a representation of liquid cloud both where it is directly diagnosed and where its presence is considered likely (e.g. within rain); however, the retrieval is not applied below a radar reflectivity threshold of $-21$ dBZ, below which sedimentation velocities are not available from C-CD (Kollias et al., 2023). This further limits the representation of non-precipitating clouds in the C-CLD product.


### 2.2.4 Synergistic retrievals


The L2b ACM-COM product (Cole et al., 2022) includes a composite atmosphere comprised of L2a ATLID and CPR products. Both A-ICE and C-CLD are used for ice cloud and snow. In volumes where A-ICE and C-CLD both contain ice cloud or snow, the product with the lowest uncertainty in retrieved effective radius and water content is used. C-CLD provides information on rain and liquid cloud. The strong lidar backscatter signal from liquid clouds being invaluable to the detection and classification of liquid cloud layers (including mixed-phase cloud) in A-TC (Donovan et al., 2023b), but no L2a ATLID liquid cloud retrieval is available due to the rapid extinction of the lidar signal. A-EBD is used for aerosols fields.


The ACM-CAP product is a unified retrieval of cloud, aerosol and precipitation from the synergy of ATLID, CPR and MSI measurements (ACM-CAP; Mason et al., 2023). One advantage of a synergistic retrieval is the capacity for a smooth transition between the parts of a cloud that are detected only by ATLID (i.e. optically thin cirrus and the tops of high ice clouds), regions that are detected by both ATLID and CPR (i.e. the tops of most optically thick ice clouds), and those detected by CPR alone after ATLID is extinguished (i.e. within physically deep and optically thick clouds, or below layers of liquid cloud). The assimilation of MSI visible radiance and thermal infrared brightness temperature channels also provides passive constraints on the solar albedo and cloud-top properties that can only be applied consistently in a unified retrieval that represents complex and layered scenes with all combinations of hydrometeors.






**Table 1.** A summary of which EarthCARE L2a & b data products can be intercompared at nadir according to their geometry and physical quantity. *Italics* indicate profiling retrieval products that are provided on higher dimensions, but can be vertically integrated or subset for comparison with other products (e.g. deriving ice water path from ice water content). Products in gray are not included in this intercomparison. Note that some quantities listed here may be derived from variables in the L2 products: for example, both C-CLD and ACM-CAP report "mass flux" in units of kg $\text{s}^{-1}$ $\text{m}^{-2}$ rather than "rain/snow rate" in units of mm $\text{h}^{-1}$; however, the conversion between these terms is unambiguous, and the terms are used interchangeably.

| | PASSIVE, LAYERWISE OR INTEGRATED (SWATH OR NADIR) | | PROFILING (NADIR) | |
| --- | --- | --- | --- | --- |
| | Physical quantity | L2 product(s) | Physical quantity | L2 product(s) |
| Macrophysics | Layer detection | M-COP, A-ALD, A-CTH, AM-CTH | Detection | A-FM, C-FMR |
| | Classification | M-CM, A-ALD, AM-ACD | Classification | A-TC, C-TC, AC-TC |
| Ice cloud & snow | Optical thickness | M-COP, *A-EBD*, *ACM-COM*, *ACM-CAP* | Extinction | A-EBD, ACM-COM, ACM-CAP |
| | Effective radius | M-COP, *A-ICE*, *ACM-COM*, *ACM-CAP* | Effective radius | A-ICE, ACM-COM, ACM-CAP |
| | Water path | M-COP, *A-ICE*, *C-CLD*, *ACM-COM*, *ACM-CAP* | Water content | A-ICE, C-CLD, ACM-COM, ACM-CAP |
| | | | Median diameter | C-CLD, ACM-CAP |
| | | | Number conc. | C-CLD, ACM-CAP |
| | Surface snowfall rate | *C-CLD*, *ACM-CAP* | Snowfall rate | C-CLD, ACM-CAP |
| Liquid cloud | Optical thickness | M-COP, *A-EBD*, *ACM-CAP* | Extinction | A-EBD, ACM-COM, ACM-CAP |
| | Effective radius | M-COP, *ACM-CAP* | Effective radius | ACM-COM, ACM-CAP |
| | Water path | M-COP, *C-CLD*, *ACM-CAP* | Water content | C-CLD, ACM-COM, ACM-CAP |
| Rain | Water path | *C-CLD*, *ACM-CAP* | Water content | C-CLD, ACM-CAP |
| | | | Median diameter | C-CLD, ACM-CAP |
| | | | Number conc. | C-CLD, ACM-CAP |
| | Surface rain rate | *C-CLD*, *ACM-CAP* | Rain rate | C-CLD, ACM-CAP |
| Aerosol | Optical thickness | M-AOT, *A-AER*, *A-EBD*, A-ALD, AM-ACD, *ACM-CAP* | Extinction | A-AER, A-EBD, ACM-CAP |
| | Lidar ratio | A-ALD | Lidar ratio | A-AER, A-EBD, ACM-CAP* |
| | Linear depol. ratio | A-ALD | Linear depol. ratio | A-AER, A-EBD |
| Radiation | TOA flux | BMA-FLX, ACM-RT | Flux | ACM-RT |
| | Surface flux | ACM-RT | | |
| | | | Heating rates | ACM-RT |



## 3 Results

For the intercomparison results presented here, both the model truth and all retrieval products are mapped onto the Joint Standard Grid using a nearest-neighbour interpolation. The intercomparison is illustrated by time series of retrieved bulk quantities (e.g. ice water content, total aerosol extinction) and/or their vertical integrals (e.g. ice water path, total aerosol optical depth). For this purpose we use the Halifax test scene, a descending frame (1/8th of an EarthCARE orbit, or around 5000 km) that passes over the mid-latitudes of the western North Atlantic (including Halifax, Nova Scotia). Of the three

simulated EarthCARE test scenes the Halifax scene samples the widest range of regimes including deep and shallow ice clouds with snow and rain, supercooled liquid and mixed-phase clouds, liquid boundary-layer clouds, and aerosols including marine sea-salt and continental pollution layers. Statistical evaluations are carried out using all three simulated test scenes.

### 3.1 Ice clouds and snow

Ice clouds, snow, graupel and hail are represented as distinct classes within the GEMS model; however, in observations there

is no clear threshold between ice cloud and precipitation particles, so it is often necessary in retrievals to treat all glaciated hydrometeors as a continuum. Nevertheless, different instruments are more sensitive to different sizes of ice particles, and this is reflected in the retrieved quantities available in different EarthCARE products. Single-instrument retrievals in the optical spectrum (i.e. ATLID and MSI) are sensitive to smaller ice clouds particles: A-ICE reports profiles of ice water content (IWC), extinction and effective radius, and M-COP reports ice water path (IWP), optical thickness and cloud-top effective radius; these

are also the quantities of interest for downstream radiative transfer modelling processor ACM-RT, and are therefore reported in ACM-COM. In the microwave spectrum (i.e. CPR) larger snowflakes and precipitating particles are dominant: therefore C-CLD reports profiles of IWC, mean volume diameter, normalised number concentration and precipitation rate $S$, which are characteristic of radar retrievals. The synergistic retrieval ACM-CAP assimilates radar, lidar and radiometer measurements and therefore reports all of the above quantities.

In this section we first intercompare the IWC and IWP, which are common to all retrievals, and then compare selected quantities relating to optical retrievals (i.e. cloud effective radius) and radar retrievals (i.e. snow rate) separately. Figure 1 shows timeseries from the Halifax scene, with GEM-model true and retrieved IWC for the active retrievals (panels a through e), and IWP for all products (panel f).

As discussed above, the lidar and radar retrievals are concerned with different parts of the size spectrum. A-ICE (Fig. 1b)

accurately resolves the features at cloud-top, including very high values of IWC up to $1\,\mathrm{g\,m^{-3}}$ in the deepest frontal part of the cloud (36–38°N), and some of the lowest values less than $1\,\mathrm{mg\,m^{-3}}$ in the optically thin anvils (38–44°N) and high-latitude clouds (65–67°N). In these thinnest clouds it is possible to distinguish between the high degree of along-track smoothing used for "weak" backscatter targets, and the more noisy or grainy retrieved values in the regions with moderate backscatter. Throughout the scene, the profile of retrieved IWC is limited by the extinction of ATLID (shaded in dark grey). Conversely, C-

CLD (Fig. 1c) accurately resolves the features of precipitating ice at high latitudes (48–66°N) and deep within the frontal cloud regime (35–45°N) with values of IWC up to $1\,\mathrm{kg\,m^{-3}}$, but misses areas of non-precipitating ice cloud in the optically thin



**Figure 1.** Ice water content from (a) the GEM model, (b) A-ICE, (c) C-CLD, (d) ACM-COM and (e) ACM-CAP, for the Halifax scene, and (f) corresponding intercomparison of ice water path including M-COP. Shading in panel b indicates where A-ICE cannot retrieve ice clouds due to the extinction of the ATLID signal; shading in panels c and d indicates profiles not retrieved by C-CLD due to multiple scattering or attenuation of CPR signal.





high and mid-level cloud layers, and in ice or mixed-phase clouds between precipitating cells (63°N) and where the shallowest cloud layers may be obscured by surface clutter (45–55°N). The greatest differences between C-CLD and the GEM model are a deficit of IWC at the top of the deep frontal/anvil cloud (37–38°N), where the high IWC values are dominated by smaller ice
cloud particles rather than the precipitating snow to which CPR is most sensitive.

The L2b products take advantage of the complementary lidar and radar measurements of different parts of the ice cloud and snow continuum to produce a more complete representation of the scene: ACM-COM (Fig. 1d) by merging A-ICE and C-CLD resolving conflicts by selecting the product with the lower retrieval uncertainty in each volume; and ACM-CAP (Fig. 1e) by carrying out a synergistic retrieval in which all ATLID, CPR and MSI measurements are used wherever they are available.
ACM-COM inherits IWC from A-ICE at cloud-top and in optically thin clouds, and from C-CLD deep within ice clouds and in snow. In many places the merged IWC fields are consistent (throughout the high-latitude clouds and in the mid-level parts of the frontal cloud) and appear seamless, while discontinuities are evident where the two products differ (in the anvil, and in mid-level clouds around 44°N and 65°N). While cloud-top IWC from A-ICE is available above the convective core around 36°N, the lower parts of the profile are not available in profiles where CPR is strongly affected by radar multiple scattering
and attenuation, and C-CLD does not currently attempt a retrieval (Mroz et al., 2023). ACM-CAP most closely resembles the GEM model IWC. Many of the features in optically thin ice clouds are accurately resolved, including values of IWC less than 1 mg m$^{-3}$: it is likely that ACM-CAP's Kalman smoother is less aggressive in applying along-track smoothing to compensate for noise in the ATLID particulate backscatter when compared with A-ICE. In heavy snowfall (64–65°N and 37°N) ACM-CAP performs as well as C-CLD, with the notable difference that ACM-CAP retrieves profiles of IWC in convective cell where CPR
is dominated by multiple scattering. Since radar multiple scattering effects were included in the instrument simulators used to generate the test scenes (Donovan et al., 2023a) and ACM-CAP includes multiple scattering in its radar forward-model (Mason et al., 2023), some information is still available to constrain retrievals in heavy precipitation. Values of IWC in these profiles reach the values of 1 kg m$^{-3}$ in the GEM model, but are under-constrained by measurements and are lower than the model truth both in the upper levels and close to the melting layer.
The timeseries of retrieved IWP (Fig. 1f) generally reflect the strengths and weaknesses of the products described above: A-ICE generally performs well in parts of the scene with low to moderate IWP, and exhibits deficits in deep and mixed-phase clouds where ATLID is rapidly extinguished; C-CLD performs well in the high-IWP profiles (with the exception of the heaviest precipitation cells, which are not retrieved) but misses or underestimates IWP in non-precipitating clouds. IWP from ACM-COM and ACM-CAP are both broadly closer to the GEM model: ACM-COM overestimates IWP in parts of the scene
dominated by optically thin ice clouds, which are retrieved by A-ICE with a high degree of along-track averaging, and has a deficit in the deep precipitation where C-CLD does not report a retrieval; ACM-CAP overestimates IWP in the shallowest boundary layer clouds (52–55°N) where CPR measurements are obscured by surface clutter, and underestimates IWP in the anvil part of the frontal regime (37°N) and the small region of optically thin ice cloud around 34°N. M-COP is available in the sunlit region of the scene equatorward of 50°N: the passive retrieval shows very strong performance in parts of the scene where
the active instruments can be limited by extinction and multiple scattering, such as the deep precipitation around 36–38°N. The





mid-level and layered cloud regimes 42–45°N are more challenging for a passive retrieval, and M-COP underestimates IWP in places.

Figure 2 presents a statistical evaluation across all three test scenes, comparing all IWP products against the GEM model. The statistical evaluation plots consist of two panels: In the lower panel of each plot, a joint histogram compares the GEM/CAMS
model truth against the retrieved quantity, where a perfect retrieval would follow the diagonal. Annotated at the top-left of each panel is the correlation coefficient $r$ and the root-mean-square (RMS) error. The frequency of occurrence colourmap is log-scaled to exaggerate rare occurrences. The top panel compares the probability density functions of the same quantity: the black lines show all data from the model truth, and the red lines show all of the retrieved data; the grey shading indicates the model truth only in volumes where the retrieval is also defined, while the red shading indicates retrieved values in volumes
where none exist in the model truth. For an omniscient observation, the grey shading would match the black line, while in a perfect retrieval the red line would match the grey shading. Annotated at the top-left of each panel is the ratio of the number of retrieved volumes to the number of volumes containing the quantity in the model, a measure of how completely the product apprehends the underlying model truth. As was evident in the Halifax scene, M-CLD retrievals are not available in all profiles, but provide a good estimate in deep and optically thick ice cloud: for IWP between 0.01 and 10 kg m$^{-2}$ the passive retrieval is
well-correlated with the GEM model ($r = 0.77$), with an RMS error of +173/-63%. The capacity of ATLID to detect optically thin ice cloud is evident in A-ICE and ACM-CAP; however, in both products the retrieved IWP is often over-estimated below around 0.001 kg m$^{-2}$; for this reason, we take 0.001 kg m$^{-2}$ as the lower threshold for the calculation of correlation coefficients and error metrics. At higher IWPs the extinction of ATLID is reflected in A-ICE, which underestimates IWP above around 0.1 kg m$^{-2}$. In contrast, C-CLD performs well at moderate to high IWPs with a strong correlation ($r = 0.91$) and an RMS error
of +262/-72%, with the caveat that profiles with the most extreme values, which tend to be subject to multiple scattering and strong attenuation, are not currently retrieved in this product. ACM-COM combines the advantages of A-ICE and C-CLD, while suppressing some extreme values: at low IWP ACM-COM resembles A-ICE, but C-CLD retrievals compensate for the extinction of ATLID at values greater than 0.1 kg m$^{-2}$, resulting in a correlation coefficient closer to that of C-CLD, and reduced random error. With the benefits of radar-lidar synergy in ice clouds, ACM-CAP is strongly correlated to the GEM
model ($r = 0.97$) with a very strong representation through moderate to high values of IWP, and an RMS error of +71/-41%. In columns with IWP below the threshold of 0.001 kg m$^{-2}$, ACM-CAP shares the challenges apparent for A-ICE and ACM-COM, being both biased high and subject to a large random error.

The effective radius of ice clouds is not easily constrained by single-instrument retrievals, which must rely on assumptions about the relation between extinction, water content and effective radius. In A-ICE ice cloud effective radius (Fig. 3b) is a
function of atmospheric temperature (Donovan et al., 2023b), which is propagated into the lidar-only portion of ACM-COM (Fig. 3c). Although C-CLD does not itself report ice effective radius, where radar information is available and has a lower retrieval uncertainty, ACM-COM derives ice effective radius based on the C-CLD retrieval (described in Cole et al., 2022); however, the differences between these two products can lead to large and noisy discontinuities in the composite field, where C-CLD retrievals result in significantly lower ice effective radii. In contrast, the synergistic ACM-CAP retrieval assimilates—at
least in parts of the profile—both the radar-lidar ratio and two thermal infrared channels from MSI, which constrain the retrieval





**Figure 2.** Ice water path (IWP) evaluation for L2a products (a) M-CLD, (b) A-ICE, (c) C-CLD, and L2b products (d) ACM-COM and (e) ACM-CAP). The top part of each panel shows probability density functions (PDFs) for all GEM model data (black line), GEM model data in volumes that are successfully retrieved by the product in question (grey shading), retrieved data (red line) and retrieved values in volumes where the GEM model does not include ice or snow (red shading). The percentage of profiles containing ice and snow in the GEM model that are reported in the product is given in the top-left of each sub-panel, as an indicator of the coverage provided by each product. The bottom part of each panel shows joint histograms of GEM model versus retrieved data, where a perfect retrieval would lie entirely along the diagonal. The colour scale is logarithmic to exaggerate rare features, and the mean and selected deciles of the distribution are overlaid. To summarise the performance of each product, the correlation coefficient $r$ and root-mean-square (RMS) error are given in the top-left of each sub-panel.



**Figure 3.** Profiles of ice effective radius from (a) the GEM model, (b) A-ICE, (c) ACM-COM, and (d) ACM-CAP for the Halifax scene. Panel (e) intercompares GEM model and retrieved cloud-top effective radius (averaged where optical thickness is less than 1). Shading in panel b indicates where A-ICE cannot retrieve ice effective radius due to the extinction of ATLID signal, and shading in panel c indicates where ACM-COM has no information due to C-CLD not retrieving in profiles strongly affected by multiple scattering or attenuation of CPR signal.



of ice effective radius more smoothly and accurately (Fig. 3d). Some errors in ACM-CAP with respect to the GEM model are still evident in parts of the Halifax scene, especially in the convective core (37°N) where the density of ice particles is poorly constrained, and at the edge of the cold rain feature (42°N) where the GEM model contains freezing rain but AC-TC diagnoses snow. M-COP retrieves cloud properties over a plane-parallel, vertically homogeneous layer. In order to intercompare cloud-

top and profiling effective radius retrievals, Fig. 3e shows cloud-top effective radius calculated from A-ICE, ACM-CAP and the GEM model by taking the average cloud effective radius where optical depth is less than 1 (note that the intercomparison is mostly insensitive to the choice of threshold). M-COP very accurately retrieves cloud-top ice effective radius in ice-only deep frontal clouds in the Halifax scene (36–38°N), but significantly underestimates those in layered parts of the scene (around 50 and 41–44°N), where the lower liquid cloud layers may dominate the radiances. This is a well-known challenge for passive

retrievals, and layered cloud scenes can be filtered out using the MSI cloud-top phase (M-CP; Hünerbein et al., 2023b).

Estimates of snowfall are limited to retrievals taking advantage of CPR, which is sensitive to the larger snowflakes, penetrates through the profile of snow, and includes Doppler velocity measurements of snow fallspeed. C-CLD and ACM-CAP report snow mass flux, a quantity which combines information about both IWC and the fallspeed of snowflakes. Accordingly, snow rates are associated with larger retrieval uncertainties related to both the assumed mass-size relation of ice particles and their

terminal velocities. The model truth used here combines the mass flux of all ice species, including graupel; however, in practice the snow rate is dominated by the unrimed snow species. Figure 4 compares snow rates for the Halifax scene both as vertical profiles and at selected levels. Both retrievals perform well, with slight differences evident in parts of the scene associated with mixed-phase cloud (e.g. 39–45°N), where ACM-CAP tends to estimate slightly lower snow rates, perhaps associated with simultaneously retrieving liquid cloud. The clearest difference between the retrievals are that ACM-CAP retrieves very light

snow rate up to the edges and tops of clouds, taking advantage of the more sensitive synergistic target classification. Further, C-CLD does not attempt a retrieval of snow rates in profiles strongly affected by multiple scattering and radar attenuation (i.e. the convective cell around 36–37°N).

An evaluation of snow rate retrievals across all test scenes is presented in Figure 5, where the combined mass flux of all ice species from the GEM model is used. C-CLD and ACM-CAP have similar correlation coefficients with GEM model

data ($r = 0.91$ and $0.97$ respectively), and the same RMS erorrs of +133/-57%. C-CLD prioritises snow, and does not report mass fluxes below around 1e-3 mm h$^{-1}$, so this is taken as the lower threshold for calculating correlation coefficients and error metrics. ACM-CAP retrieves snowfall rates in ice clouds detected only by ATLID or where CPR's Doppler velocity measurements provide only a weak constraint on vertical motion, with reported snow rates down to 1e-5 mm h$^{-1}$. ACM-CAP covers 63% of volumes containing ice cloud and snow, and C-CLD around 28%; however, we note that the lighest mass fluxes

retrieved by ACM-CAP still overestimate those in the ice clouds of the GEM model, which likely reflects different assumptions about the structure and terminal fallspeeds of the smallest ice particles.

ACM-CAP also reports snow rates in the heaviest precipitation; however, in contrast to the integrated quantity (IWP; Fig. 1), there is a tendency to underestimate the heaviest snow rates around 5 mm h$^{-1}$ and above due to the the onset of attenuation, multiple scattering, and associated uncertainties in the presence of supercooled liquid and the degree of riming, especially in

convective cells.

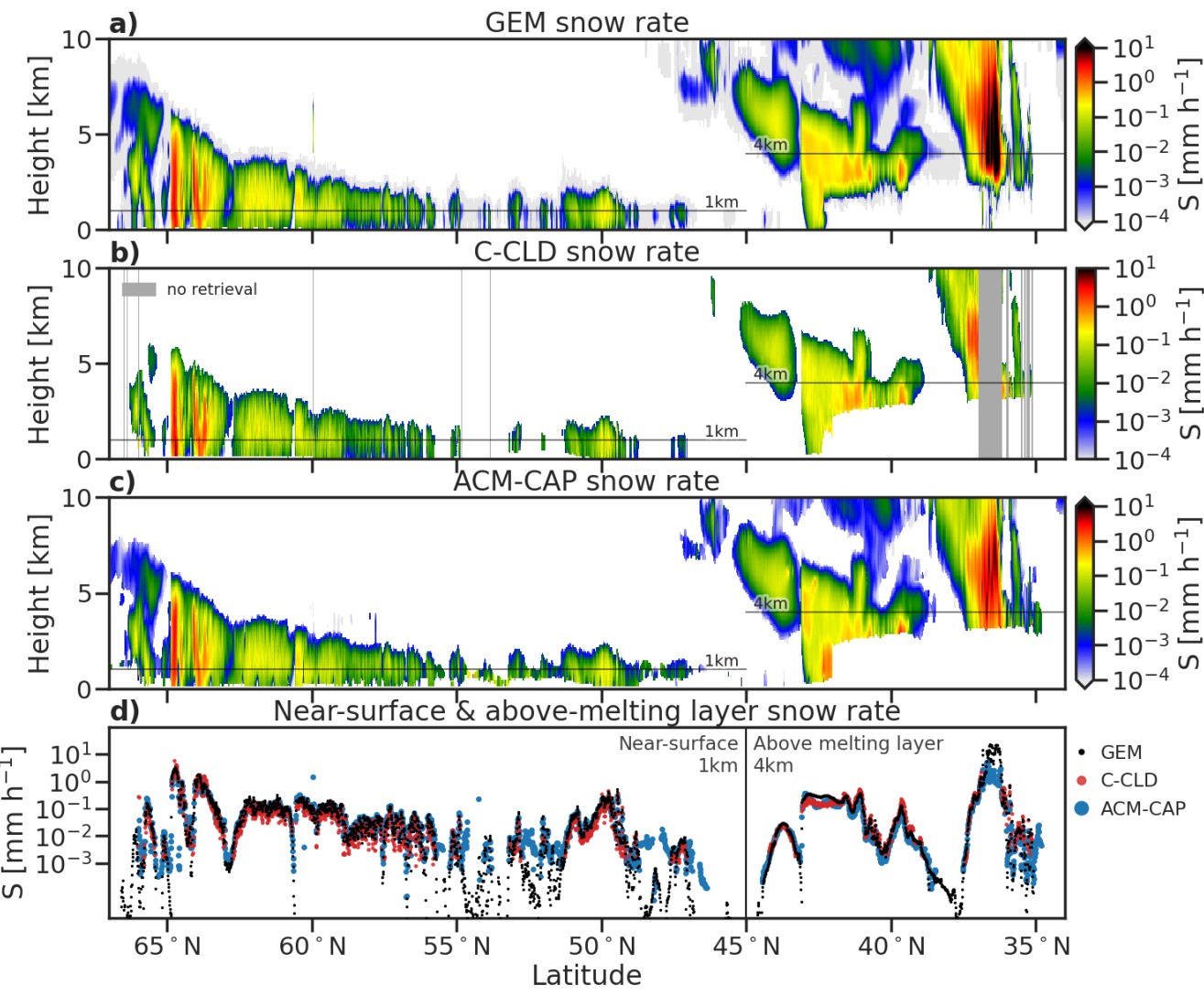

**Figure 4.** Snow rate from (a) the GEM model, (b) C-CLD, (c) ACM-CAP, and (d) intercompared at selected levels over the Halifax scene. Shading in panel b indicates where C-CLD does not retrieve profiles where CPR is affected by multiple scattering and/or attenuation.





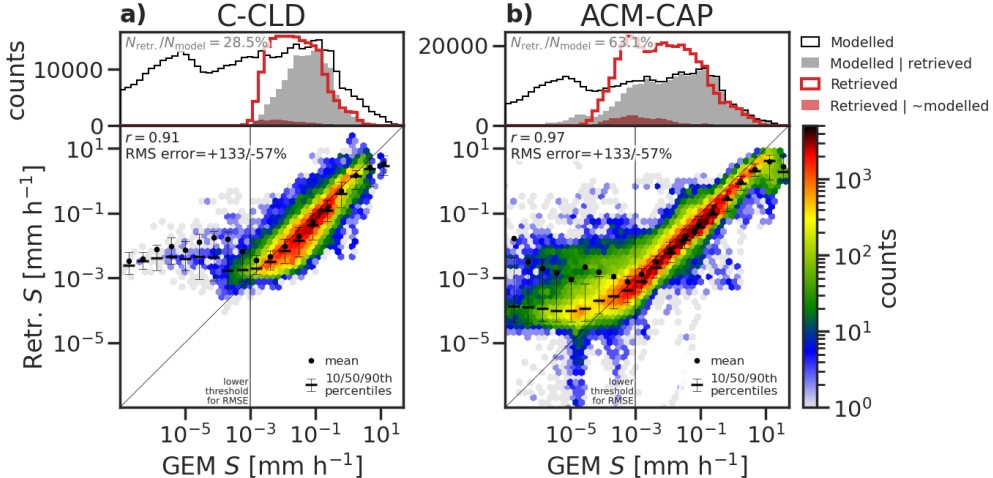

**Figure 5.** As for Fig. 2, but for snow rate (i.e. ice mass flux) from (a) C-CLD and (b) ACM-CAP for all test scenes against that from the combination of all ice species in the GEM model.

### 3.1.1 Evaluation of retrieval uncertainties

Ice clouds and snow are the one part of the atmosphere in which measurements from the radar and lidar overlap usefully for synergistic retrievals; however, the thinnest ice clouds in the test scenes are only partially detected by ATLID. The A-FM (van Zadelhoff et al., 2023b) and A-PRO (Donovan et al., 2023b) processors use multiple techniques to distinguish cloud features
from the ATLID signal, including along-track averaging to distinguish thin ice clouds from aerosols. In an intercomparison of target classification products, Irbah et al. (2023) showed that around 10% of the ice clouds in the test scenes are seen by ATLID but not detected by CPR. These cirrus and cloud-top features detected only by lidar contribute almost negligibly to the total mass of ice in the tests scenes, which is dominated by snow, but will nevertheless be radiatively important, both for the assimilation of MSI radiances in ACM-CAP and in the broadband radiative closure assessment. ACM-CAP has employed a
method of extracting some synergistic information in lidar-only parts of the ice cloud by ensuring that the forward-modelled radar reflectivity in these regions does not exceed the threshold of CPR noise (Mason et al., 2023).

Ice clouds and snow seen by the radar only are dominated by precipitation in deep clouds, where ATLID becomes extinguished near cloud-top, but will also include any ice clouds obscured by the extinction of the lidar in other contexts, such as by layers of supercooled or mixed-phase cloud aloft.
Fig. 6 compares IWC retrievals from the single-instrument and composite retrievals against ACM-CAP in the distinct lidar-only, synergistic, and radar-only parts of the ice clouds and snow in the three test scenes. The IWC here is the combined mass of all ice, snow and graupel species from the GEM model.

In ice clouds detected only by lidar, the distribution of IWC in the GEM model is bimodal with peaks at around 1e-7 kg m$^{-3}$ and 1e-6 kg m$^{-3}$ (grey shading in the top panel of Fig. 6a). These distinct peaks may relate to smaller ice particle and larger



**Figure 6.** As for Fig. 2, but an intercomparison of IWC retrievals from single-instrument and composite products (a-c) and ACM-CAP (d-f), distinguishing between ice clouds and snow that are detected by lidar-only (a & d), both radar and lidar (b & e), and radar only (c & f).





snowflake species within the GEM model. This bimodality is also resolved by A-ICE, which distinguishes between strong and weak ice cloud targets, and where the retrieval of ice from weak targets are subject to along-track averaging (Donovan et al., 2023b). ACM-CAP includes no such distinction between ice species, and does not resolve a bimodal distribution of IWC in the lidar-only part of the cloud (red line in top panel of Fig. 6d). ACM-CAP better resolves the upper end of the IWC in this part of the cloud, while A-ICE is biased high. While both A-ICE and ACM-CAP have difficulty retrieving IWC below 1e-6 kgm$^{-3}$

as the limits of ATLID signal-to-noise ratio are reached, it is notable that the along-track averaging applied in processing the ATLID measurements enables more accurate retrievals of low IWC down to below 1e-8 kgm$^{-3}$ in A-ICE, while ACM-CAP appears to saturate below around 1e-7 kg m$^{-3}$.

Where radar and lidar measurements are available, ACM-COM selects from A-ICE and C-CLD the product with the lowest estimated uncertainty in each volume. As a result, the random error in ACM-COM is exaggerated in this region, with frequent

values varying over an order of magnitude, where the transition between A-ICE (biased high) and C-CLD (biased low) lead to discontinuities. Exaggerating the differences between products, this is the region where ACM-CAP benefits the most from radar-lidar synergy. ACM-CAP has a correlation coefficient of 0.93 and an RMSE of +71/-41%, compared with $r = 0.67$ and RMSE=+194/-66% from ACM-COM.

For ice clouds and snow detected only by radar, we compare C-CLD and ACM-CAP. C-CLD covers a narrower range of

IWC, as it does not include profiles subject to radar multiple scattering and does not retrieve IWC below 1e-7 kg m$^{-3}$. The lower limit may be due to C-CLD completing retrievals only where there is sufficient SNR for a Doppler velocity measurements, which results in a lower limit of CPR radar reflectivity around -21 dBZ. Both C-CLD and ACM-CAP appear to have a slightly low bias in IWC, of a similar degree, which is greatest through the most frequent values of IWC around 1e-4 to 1e-3 kg m$^{-3}$. C-CLD and ACM-CAP have high to very high correlation coefficients (0.85 and 0.91, respectively). We note that

when stratifying this evaluation by temperature (not shown) the correlation coefficient and RMSE of C-CLD degrades toward the colder temperature ranges dominated by ice particles rather than snowflakes. This reflects choices made in the development of C-CLD to prioritise the representation of snow (Mroz et al., 2023). ACM-CAP, with the benefit of lidar measurements in ice clouds, has roughly similar correlation coefficients and RMSE across all temperature ranges.

Previous evaluations of active retrievals of ice and snow have used airborne in-situ measurements to quantify the uncertainty

in retrieved IWC. Hogan et al. (2006) estimated a range of +55/-35% in IWC for cloud radar retrievals in ice clouds and snow between $-20°C < T < -10°C$ that were clear of liquid droplets, rising to +90/-47% for ice clouds with $T < -40°C$. These uncertainties were based on variations in particle size distribution, although the presence of mixed-phase cloud was observed to lead to substantial biases in retrieved ice water content. Delanoë and Hogan (2008) estimated the error in the retrieved extinction to be around 20 to 40% in ice clouds retrieved with radar-lidar synergy, and around 50% in the radar-only region.

The simulated test scenes allow us to quantify uncertainties in retrieval of IWC in a way that includes additional uncertainties, both in the simulated instrument noise and based on the cloud properties represented in the GEM model, such as: variations in the density of ice particles, the presence of supercooled liquid, and the possibility of riming in mixed-phase clouds and convective cores, as well as the simulated instrument noise. The retrieval uncertainties presented here are around twice that presented in other studies, which we attribute to the inclusion of these additional sources of uncertainty. Comparing ACM-



CAP retrievals in the parts of the scene detected with radar-lidar synergy and radar only, we note a similar doubling of the uncertainty in the radar-only only part of the cloud to that estimated in Delanoë and Hogan (2008). An updated evaluation and intercomparison of EarthCARE's single-instrument and synergistic ice cloud and snow retrievals using in-flight data will be made as a part of calibration/validation activities.

## 3.2 Liquid cloud

The attenuation of ATLID is so rapid in liquid clouds that no single-instrument ATLID retrieval of their properties is attempted; however, A-EBD does report ATLID extinction coefficients in volumes identified as liquid cloud. The highly sensitive CPR may be capable of detecting non-precipitating liquid clouds in some cases, and C-CLD includes retrievals of liquid water content (LWC) from non-precipitating liquid clouds (Mroz et al., 2023); however, the simulated test scenes do not include sufficient sampling of stratocumulus regimes to demonstrate in this intercomparison. The intercomparison of liquid-cloud retrievals for

the Halifax scene (Figure 7) includes the passive retrieval (M-COP), the radar retrieval (C-CLD), and the synergistic product (ACM-CAP); the representation of liquid cloud in the composite atmosphere of ACM-COM is inherited entirely from C-CLD, so is not evaluated separately.

C-CLD accounts for the likely presence of liquid cloud water in the presence rain, using an adiabatic distribution from the melting layer to the lifting condensation level (Mroz et al., 2023); this liquid cloud contributes to CPR's path-integrated attenu-

ation (PIA) measurement even when the radar reflectivity is dominated by larger precipitating drops. ACM-CAP makes similar assumptions in the presence of rain, rimed snow and heavy precipitation, where a simplified profile of liquid water content can be constrained by both PIA and MSI visible radiances in daylit volumes (Mason et al., 2023). The inferred likelihood of liquid cloud colocated with rain is evaluated for the three test scenes from a target classification perspective in Sect. 6.2 of Irbah et al. (2023). Comparison with LWC in the GEM model (Fig. 7a–c) suggests that these are justified, if coarse, assumptions when

applied in the stratiform cold rain part of the Halifax scene (40–43°N): both C-CLD and ACM-CAP retrieve values of LWC around $0.001$ kg m$^{-3}$ in these areas, close to that in the GEM model.

Elsewhere, only ACM-CAP retrieves liquid cloud, both by synergy with ATLID and by making more aggressive assumptions of liquid cloud through mixed-phase areas. In mixed-phase cloud above this region, ACM-CAP benefits from the detection of liquid cloud by ATLID as well as the diagnosis of rimed snow by CPR, which are both included in the synergistic target

classification (AC-TC); retrieved values of LWC are close to the GEM model truth in the elevated mixed-phase layer, but under-estimated near the melting layer. In the deeper precipitation (36–38°N) liquid cloud in ACM-CAP is based on the "heavy precipitation" classes diagnosed where CPR experiences multiple scattering and strong attenuation. In these latter regions the GEM model truth shows much more complex spatial distributions of liquid cloud, and the coarse assumptions of retrieving liquid cloud in these contexts are more prone to error; nevertheless, ACM-CAP appears to include liquid cloud in roughly the

right amount.

In the sub-tropical boundary-layer clouds, ATLID measurements in synergy with MSI visible radiances appear to be a very good constraint on LWC and liquid water path (LWP) (Fig. 7d); in contrast, the discrimination of mixed-phase clouds within the high-latitude boundary layer clouds is much more challenging, and the ACM-CAP liquid water content is frequently





interrupted by gaps in the target classification. Furthermore, the retrieved values of LWP tends to be biased low in this region,
presumably both due to the difficulty in attributing ATLID measurements to ice and liquid cloud, and due to the lack of MSI
solar radiances in this part of the Halifax scene.

The passive MSI retrievals of liquid cloud (M-COP) are limited in their coverage to day-lit pixels, but exhibit very strong
performance in unambiguous cloud scenes: both in mid-latitude (45–50°N) and sub-tropical (30–35°N) boundary-layer clouds
and the single-layered or deep frontal clouds (35–41°N), the M-COP LWP is very close to the GEM model truth. The exceptions
are in very challenging layered clouds scenes (42–45°N), where LWP tends to be over-estimated.

An evaluation of M-CLD, C-CLD and ACM-CAP retrievals of LWP across the three test scenes is shown in Figure 8. M-CLD
retrievals are available in profiles with unambiguous liquid and supercooled liquid cloud layers—and only during daytime—
with a coverage of around $43\%$ of cloudy columns at nadir. Where a retrieval is returned, M-CLD is highly correlated with the
GEM model truth ($r = 0.91$), with an RMS error of +166/-62%.

C-CLD liquid cloud retrievals are limited to profiles containing rain, at least in the test scenes in which no significant
stratocumulus clouds are sampled: this amounts to around 16% of columns containing liquid clouds across the three test
scenes. The retrieved LWP is close to the GEM model LWP at the upper end of the distribution, but the moderate correlation
($r = 0.54$) and large RMS error (+630/-86%) reflect the high degree of spread through moderate values of LWP

The synergistic ACM-CAP retrievals cover the full range of LWP values in the GEM model, and reports retrievals in $96\%$ of
profiles containing liquid cloud. Considering the weak observational constraints on LWP in many parts of the test scenes, the
retrieval is strongly correlated ($r = 0.81$), with an overall RMS error of values above 1e-3 kg m$^{-2}$ is +300/-75%. The closest
correlation and least random error is at at the largest values of LWP above 1e-1 kg m$^{-2}$, with a slight positive bias through
moderate values 0.001 to 0.1 kg m$^{-2}$.

## 3.3 Rain

As with snow, the only rain retrievals are those constrained by CPR measurements: C-CLD and ACM-CAP. The greatest
differences between the retrievals are their coverage: C-CLD takes the more conservative approach of reporting rain retrievals
only where the microphysical properties are well-constrained (i.e. not in the melting layer, which is shaded in dark grey in
Fig. 9b), or where the signal-to-noise ratio is sufficient to extract a mean Doppler velocity estimate (i.e. radar reflectivity
greater than $-21$ dBZ; Kollias et al., 2023), and where a complete profile of radar measurements can be used (i.e. where
multiple scattering and strong radar attenuation are not detected, shaded in lighter grey in Fig. 9b). Since C-CLD does apply
a mass-flux continuity constraint across the melting layer, a simple interpolation of mass flux and water content quantities
is provided between the lowest volume containing ice and the highest rain in the C-CLD product (Mroz et al., 2023). In
contrast, ACM-CAP prioritises continuity and retrieves rain throughout the melting layer by accounting for discrepancies with
an increased radar forward-model errors at near-freezing temperatures. Where no mean Doppler velocity measurements are
available to constrain the raindrop size, ACM-CAP will still retrieve rain based on a-priori information—and will report a
greater retrieval uncertainty accordingly. To facilitate rain retrievals even in heavy precipitation, radar multiple scattering is

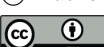

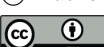

**Figure 7.** Liquid water content profiles for the Halifax scene from (a) the GEM model, (b) C-CLD, and (c) ACM-CAP. Panel (d) compares the liquid water paths retrieved by C-CLD, ACM-CAP and M-COP at the nadir pixel. Shading in panel (b) indicates that C-CLD does not retrieve profiles that are strongly affected by multiple scattering and/or extinction in the CPR signal: liquid water is only retrieved where colocated with rain, and not the melting layer.





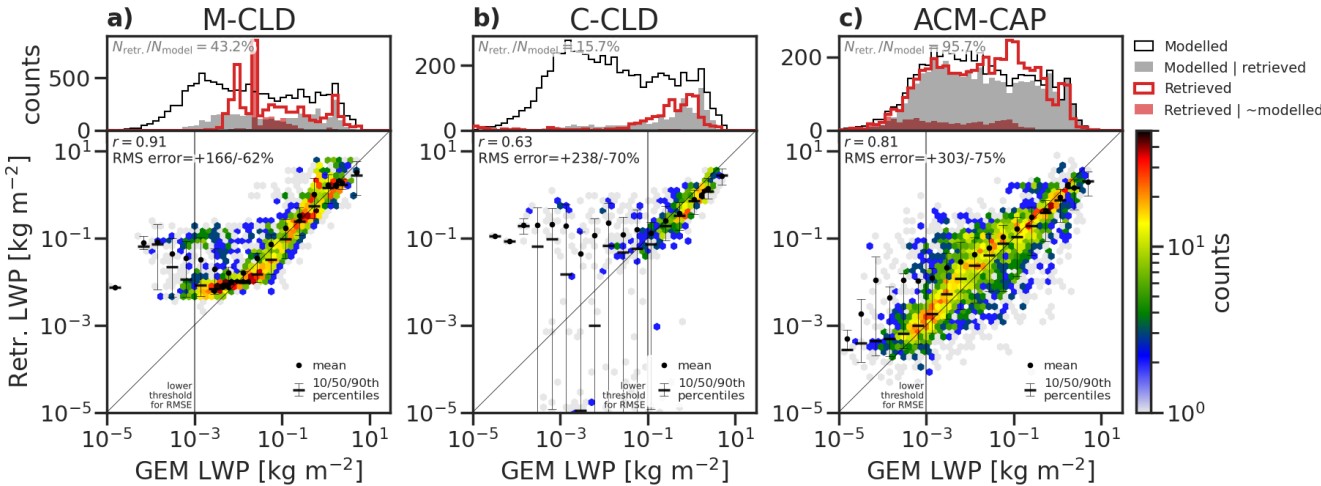

**Figure 8.** The same as Fig. 2, showing retrieved LWP for (a) M-CLD, (b) C-CLD and (c) ACM-CAP across all test scenes.

included in the ACM-CAP radar forward-model, and rain is retrieved even in conditions of complete extinction of the radar beam.

In terms of rain water content, the values retrieved by C-CLD (and included in ACM-COM) and ACM-CAP are all very close
to the GEM model truth, with the exception of ACM-CAP over-estimating rain water content in some of the light stratiform rain (40.5–42°N) part of the Halifax scene. The rain water path (RWP; Fig. 9e) shows the importance of resolving the depth of the rain layer: the C-CLD and ACM-COM products tend to under-estimate the total amount of rain water due to not including rain in the melting layer and/or surface clutter.

This is further reflected in the evaluation of RWP over the three test scenes (Fig. 10). ACM-CAP and ACM-COM each exhibit
high correlations ($r = 0.89$) and moderate RMS errors (+145/-59%); however, there are significant differences in the degree of coverage of rainy profiles. While ACM-CAP covers around 97% of rainy volumes, rain in C-CLD does not currently include the melting layer (including virgae) or heavily precipitating profiles, limiting the coverage to 46% across the three scenes. This is further reduced to 36% for ACM-COM, where the surface clutter is also not included in the current configuration.

A quantity of more immediate interest than the rain water path is the rain rate, which includes information about the fallspeed
of raindrops and hence their size distribution. The GEM model and retrieved rain rates ($R$) through a subset of the Halifax scene are shown in Fig. 11. In the stratiform cold rain (39–43°N) rain rates are between 0.1–1.0 mm h$^{-1}$ with embedded regions of heavier rain up to 10 mm h$^{-1}$; in the heaviest convective rain (36–37°N) rain rates are between 10 and 30 mm h$^{-1}$. Where both products report rain rates, they are each very close to the GEM rain rate, with some exceptions: both products are unable to retrieve rain rates under the temperature inversion (42–42.5°N) due to a limitation of the target classification (Irbah et al.,
2023), and in the lightest stratiform cold rain (40–42°N) some features are not resolved, especially close to the surface. In the heaviest convective precipitation where no radar reflectivity is available and PIA is saturated, ACM-CAP retrieves heavy rain broadly in the right magnitude, but the strong vertical gradient of rain rate is the result of a poorly-constrained retrieval.





**Figure 9.** Profiles of rain rate from (a) GEM model, (b) C-CLD, (c) ACM-COM, and (d) and ACM-CAP for the Halifax scene. Rain water paths are compared in panel e. Shading in panel (b) indicates that C-CLD does not retrieve profiles that are strongly affected by multiple scattering and/or extinction of the CPR signal, and all melting snow is treated separately from the rain retrieval.



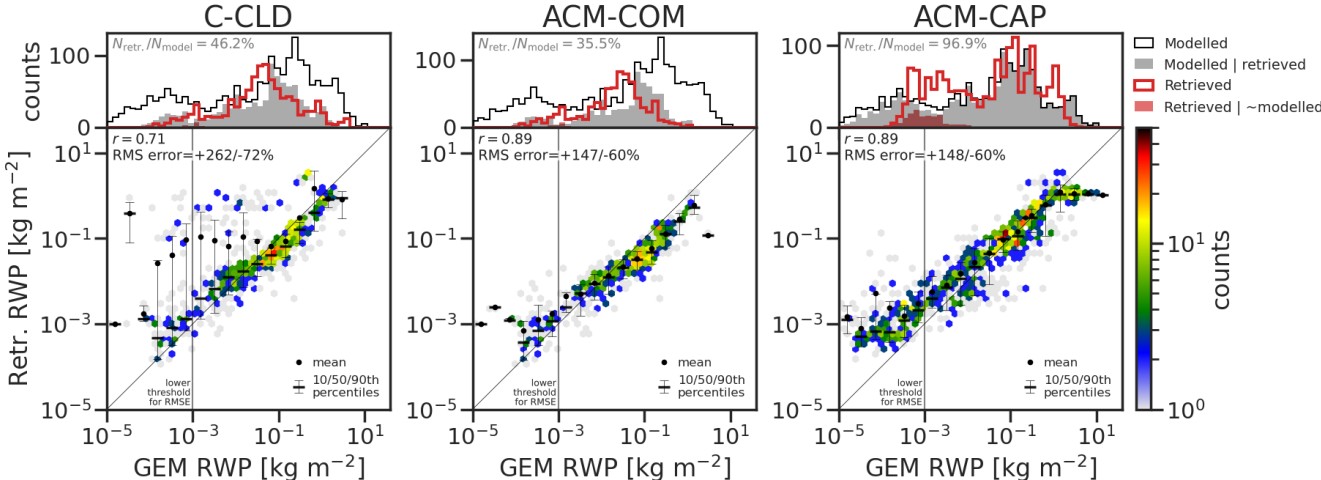

**Figure 10.** The same as Fig. 2, showing RWP retrievals from (a) C-CLD, (b) ACM-COM, and (c) ACM-CAP across all test scenes.

This may be addressed by modifying the a-priori assumptions or spatial constraints applied in profiles identified as heavy precipitation (Mason et al., 2023). The rain rate at 1km (Fig. 11d) shows that both products broadly represent the near-surface

rain rate, even resolving narrow precipitation features.

An evaluation of rain rate across the three test scenes is shown in Figure 12. The PDFs of rain rate (top panels) show that the span of the retrieved values (red line) matches very closely the GEM model values that are detected by the instrument (grey shading), and that the coverage of the ACM-CAP retrieval includes almost all moderate to heavy rain rates while missing some of the weakest features (cf. Sect. 6.3 of Irbah et al., 2023), covering around 78% of rainy volumes overall. C-CLD's

more cautious approach to the melting layer reduces coverage at all rain rates as well as the heaviest rain rates, and covers around 25% of rainy volumes from the three simulated test scenes. The joint histograms (lower panels) show that C-CLD and ACM-CAP perform very well through moderate to high rain rates ($r = 0.94$ and $0.91$, respectively), with a positive bias in ACM-CAP through moderate to low values. Both products tend toward high biases at the lowest rain rates ($R < 0.01\,\mathrm{mm\,h^{-1}}$). The larger RMS error in ACM-CAP (+184/-65%, compared to +101/-50% for C-CLD) may be the result of its less cautious

approach to screening out rainy volumes, such as volumes where mean Doppler velocity measurements are unavailable due to measurement noise.

### 3.4    Aerosols

We compare two ATLID retrievals of aerosol extinction: A-AER and A-EBD, and the aerosol quantities reported by the synergistic retrieval product ACM-CAP, which also includes some information from MSI solar radiances. The layerwise aerosol

product A-ALD also provides two estimates of total AOT, comparing the column total and the sum of identified aerosol layers. The passive MSI aerosol optical thickness retrieval from M-AOT can also be compared against the ATLID products the nadir pixel. Figure 13 shows the GEM model and retrieved aerosol extinction and total aerosol optical thicknesses (AOT) for the





**Figure 11.** GEM model (a), C-CLD (b) and ACM-CAP (c) profiles of rain rate for the Halifax scene. A comparison of GEM model and retrieved rain rates at 1 km are shown in panel d. Shading in panel (b) indicates that C-CLD does not retrieve profiles that are strongly affected by multiple scattering and/or extinction of the CPR signal, and all melting snow is treated separately from the rain retrieval.



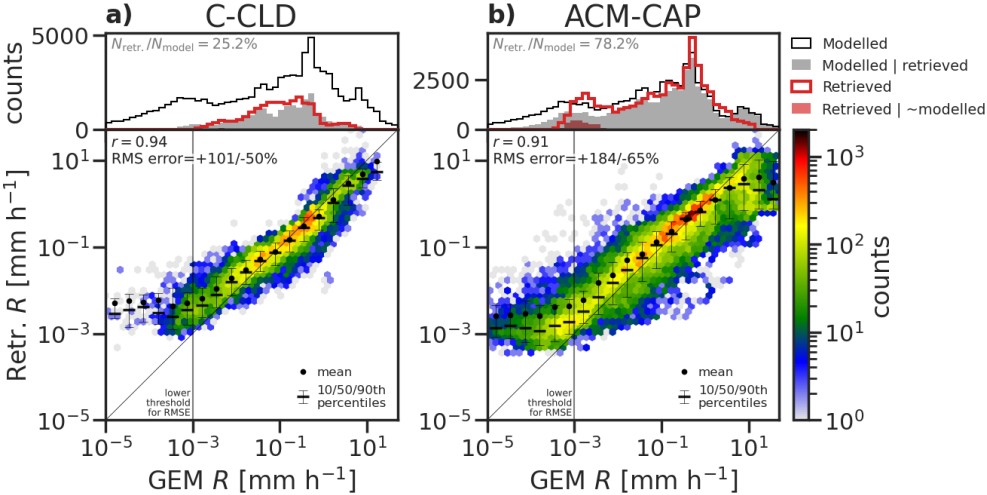

**Figure 12.** The same as Fig. 2, showing rain rate retrievals from (a) C-CLD and (b) ACM-CAP across all test scenes.

Halifax scene. Note that aerosols can only be detected and retrieved in the absence of other targets and before the ATLID signal is extinguished (shaded in the figure): clouds obscure much of the aerosols through the mid- and high-latitudes in this

scene. In the sub-tropics a layer of marine aerosols near the surface is interrupted by scattered non-precipitating cumulus clouds, and overlaid by a layer of continental pollution Irbah et al. (see target classification in Fig. 9 in 2023). An elevated layer of aerosol over the high-latitude part of the scene is largely unresolved by ATLID.

The profiling retrievals of aerosol extinction (Fig. 13b–d) broadly represent the spatial structure of aerosol extinction in the numerical model through the sub-tropical part of the Halifax scene, with some significant differences due to the different

methods used to compensate for the high degree of ATLID measurement noise when sampling aerosols. A-AER uses a 150km along-track averaging window to increase the signal-to-noise ratio for aerosols, resulting in smooth retrieval very similar to the CAMS model truth—especially in the top-most layer of continental pollution, which is not interrupted by liquid clouds or breaks in the detection of aerosols. A-EBD retains the most small-scale features in the retrieved aerosol extinction, especially notable in the vertical dimension. Small-scale local maxima in the ACM-CAP aerosol extinction result in a "dappled" texture

to the retrieved field, likely due to interruptions to the Kalman smoother around the edges of features. The removal of these artefacts will the subject of ongoing work.

Very high values of aerosol extinction near the surface in A-EBD and A-AER (e.g. 25 to 31°N and around 34°N) are likely related to contamination by liquid clouds embedded in the marine aerosol layer, which will affect both the ATLID Mie and Rayleigh backscatter channels. The number of volumes affected by this is partly reduced by using the A-TC target classification

to select for known aerosol classes: the classifications included in the A-AER and A-EBD products include greater frequencies of volumes assigned an aerosol class, but where the particulate signal is strongly affected by attenuation. Other challenging volumes, which are also better screened for by A-TC and the synergistic AC-TC classification used by ACM-CAP, occur within the mixed-phase layers between 39 and 44°N. ACM-CAP is less strongly affected by the issue of extreme aerosol extinctions





below the embedded liquid clouds, possible due to simultaneously retrieving liquid cloud and aerosols from the synergy of
both ATLID backscatter channels and MSI solar radiances, and as a consequence of making the extinction-to-backscatter ratio
a fixed property of each aerosol class, which over-constrains the aerosol extinction. Small areas of erroneous aerosol retrievals
in A-EBD and A-AER are also evident within the optically thin ice clouds (39 to 48°N), where the ATLID target classification
is based on along-track averaging of weak lidar signals. In this region ACM-CAP benefits from using the synergistic target
classification (AC-TC), wherein CPR measurements are used to clarify that no aerosol retrieval should be carried out. In high-
latitude parts of the Halifax scene ACM-CAP retrieves erroneously high near-surface aerosol extinctions, perhaps due to this
complementary information not being available: these are night-time profiles where MSI solar radiances are not available, and
within the CPR surface clutter where the radar cannot clarify ambiguities in the target classification. By visual inspection,
A-AER appears to best represent the smoothly-varying features in the large-scale and uninterrupted aerosol fields from the
CAMS model; however, in layered and complex scenes the advantages of a synergistic retrieval are evident.

When interpreting the total AOT in the Halifax scene (Fig. 13e), the challenges discussed above are still evident: in the
largely cloud-free parts of the sub-tropical aerosol scene (23-25°N and 31-33°N) A-EBD, A-ALD and ACM-CAP are closest
to the CAMS model, or biased slightly high (up to around 30% more than the model truth); conversely, A-AER underestimate
the total AOT by as much as 30%. The passive retrieval (M-AOT) is available in only the completely cloud-free profiles, and
is very close to total AOT (at 670nm) from the model truth especially in largest cloud-free region from 30–33°N. Equatorward
of 25°N, M-AOT exhibits a higher degree of noise, possibly due to small amounts of cloud near the surface. The AOT in
this region underestimates that from the model truth by around 15 to 20%. Where liquid clouds are embedded in the marine
aerosol layer (25–31°N) M-AOT cannot retrieve aerosols, and the active retrievals exhibit a much higher degree of noise. Here
ACM-CAP stays closest to the model truth (within around 50%), while both A-AER and A-EBD are biased high by as much
as 100%.

An evaluation of total AOT across all three test scenes is shown in Figure 14. In order to increase the sampling of M-AOT,
which can only report aerosol retrievals in daylit and cloud-free pixels, the M-AOT evaluation includes off-nadir data across
the entire MSI swath. The AOT at 670nm is used here, which is distinct from the 355nm AOT used for the lidar retrievals.
Across the three test scenes, M-AOT reports aerosol retrievals in 27% of pixels that contain aerosols in the model truth: mostly
this is due to profiles obscured by clouds. M-AOT exhibits a moderate correlation coefficient ($r = 0.53$) and relatively low
RMS error (+61/-38%), indicating strong performance in unambiguously cloud-free profiles.

Where there is a strong aerosol signal (AOT > 0.05), A-AER, A-EBD, A-LAY and ACM-CAP all show joint-histograms
close to the diagonal, and correlation coefficients range from moderate (ACM-CAP with $r = 0.62$ to high A-ALD with $r = 0.85$). A-AER has a higher degree of RMSE and is biased low, both consistent with its high degree of spatial smoothing. A-EBD
is more sensitive to small-scale features in ATLID's particulate backscatter measurements, reducing the RMSE but biasing the
AOT high overall. We note that the performance of A-AER and A-EBD are highly sensitive to the aerosol classifications used
to select the volumes to be included in the evaluation: when the aerosol classifications provided within each product are used
instead of that from A-TC, the random error increases (by around 50 percentage points) and correlation coefficients decrease by
0.1 to 0.2. The total columnar AOT from A-ALD is derived from the extinction profile of A-EBD, but applies a more stringent





**Figure 13.** Profiles of total aerosol extinction from (a) the CAMS model (a), (b) A-AER, (c) A-EBD and (d) ACM-CAP, and (e) the total aerosol optical thickness from A-EBD, A-AER, A-ALD and ACM-CAP (at 355nm) and the passive M-AOT (at 670nm), for the Halifax scene. Shading over the A-EBD, A-AER and ACM-CAP profiles indicates where aerosol retrievals are not available due to the complete extinction of ATLID signal, and the presence of other targets.





screening for cloudy volumes: this discards a further 25% of profiles across the three test scenes, and improves the correlation
coefficient ($r = 0.85$) and roughly halves the RMSE. ACM-CAP is notably the least biased of the aerosol products, exhibits a
moderate correlation coefficient ($r = 0.62$) that may be affected by the "dappled" artefacts evident in the Halifax scene, and
has a similar RMSE to A-EBD (+101/-50%). The performance of ACM-CAP may be due to the contribution of synergistic
observational constraints and fixed values of extinction-to-backscatter ratio. Despite the synergistic target classification helping
to screen out ice identified as aerosols, ACM-CAP is the least cautious of the aerosol products, and reports aerosols in 67% of
profiles.

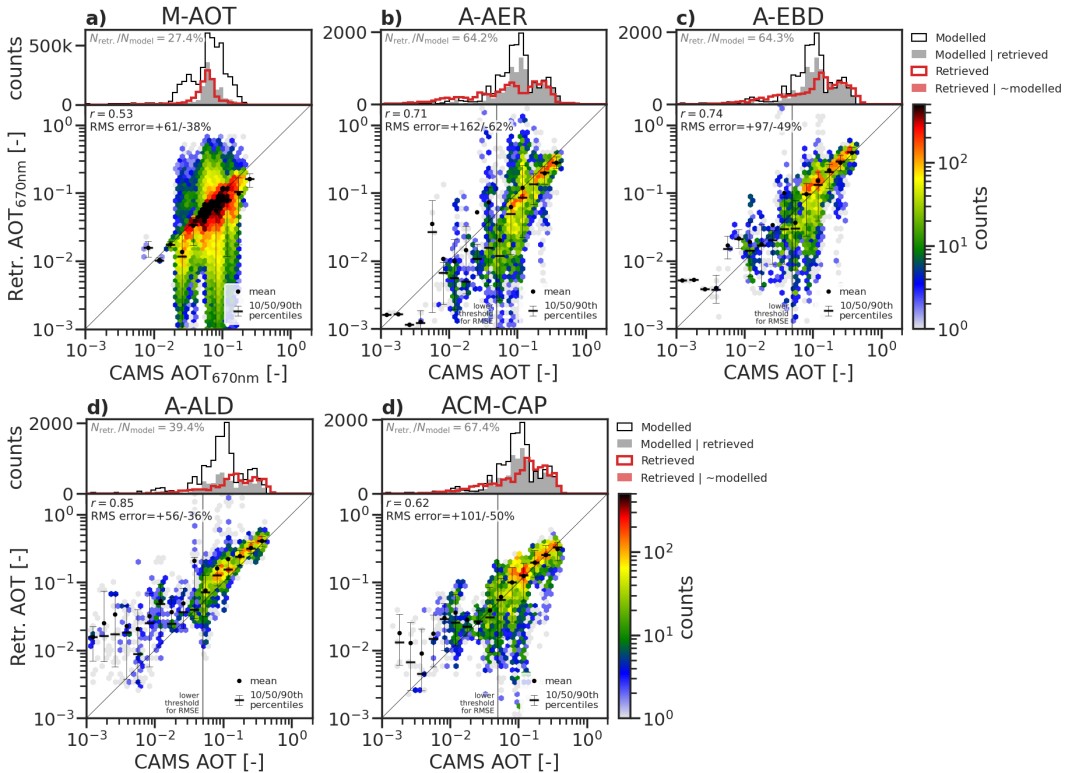

**Figure 14.** The same as Fig. 2, showing retrieved total aerosol optical thickness (AOT) from (a) M-AOT at 670nm, and (b) A-AER, (c)
A-EBD, (d) A-ALD, and (e) ACM-CAP at 355nm, across all simulated test scenes. Due to the need to avoid contamination from cloud in
aerosol-only profiles and increase the number of samples, the M-AOT evaluation (a) uses data from across the MSI swath rather than just
nadir pixels, drastically increasing the number of samples used in this panel.

## 4 Discussion and conclusions

The ESA production model for EarthCARE data products includes single-instrument (L2a) and synergistic (L2b) retrievals
of cloud, aerosol, and precipitation. This intercomparison provides an illustration of the available L2a and L2b geophysical




retrieval products, and demonstrates their capabilities and performance as applied to the simulated test scenes that have been
produced ahead of the launch of EarthCARE. An overview of the quantities included in each product is provided in Table 1.

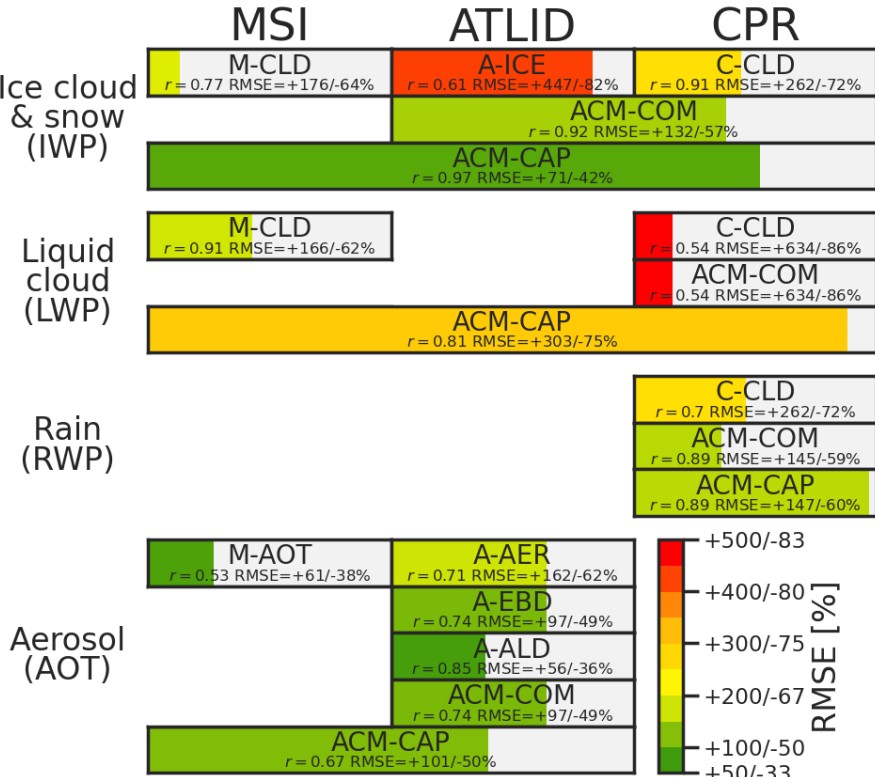

**Figure 15.** A summary of the evaluation of EarthCARE geophysical retrieval products presented in this study, based on the three simulated
EarthCARE test scenes. The columns indicate which of the EarthCARE instruments contribute to the retrieval of ice cloud & snow, liquid
cloud, rain and aerosols (with constituents grouped vertically). Each cell is coloured by the RMSE of the bulk quantities pertaining to each
class (i.e. IWP for ice cloud and snow, LWP for liquid cloud, RWP for rain, and AOT for aerosols) relative to the GEM/CAMS model truth,
and are filled according to the fraction of profiles in the model truth that are successfully reported in the product.

Figure 15 summarises the results of the evaluation of selected bulk quantities of ice cloud and snow, liquid cloud, rain and
aerosols from the L2 EarthCARE products presented here. The products are organised by the EarthCARE instruments that
contribute to their retrieval, and each cell is coloured by the RMSE of the retrieval and filled according to the percentage of
profiles that are successfully retrieved.

The benefits of instrument synergy are particularly evident in ice clouds and snow, where the ACM-COM and ACM-CAP
products cover the wide span from optically thin ice clouds detected only by ATLID, to rimed snow deep within precipitating
clouds. The composite of A-ICE and C-CLD performed nearly as well as the synergistic retrieval ACM-CAP in terms of IWP;
however, in the regions detected by both lidar and radar, the transition from A-ICE to C-CLD can manifest as discontinuities





in the profiles of retrieved ice water content and ice effective radius. A comparison distinguishing the lidar-only, synergistic and radar-only parts of the ice cloud and snow confirmed that the greatest differences between ACM-COM and the composite of single-instrument retrievals (ACM-COM) are in the ice clouds detected by both radar and lidar; however, in both the lidar-only (ice cloud) and radar-only (snow), ACM-CAP also shows somewhat higher correlations and lower RMS errors. In the lidar-only region, this may reflect the contributions of MSI radiances, as well as the pseudo-synergistic constraint provided by the absence of radar measurements. In the radar-only snow retrievals there may be some small contribution from MSI solar radiances in optically thick cloud, but we expect the majority of the differences here are due to different microphysical and microwave scattering assumptions in ice, which must ultimately be validated with in-flight EarthCARE data.

In some areas the synergistic retrieval relies almost entirely on a single instrument, and L2a products perform equally well. For both snow and rain, the C-CLD and ACM-COM retrievals both perform very strongly, with the greatest differences being in the spatial coverage, where ACM-CAP benefits from synergistic detection of ice clouds and a less cautious approach to screening out volumes at or beyond the limitations of the instruments to detect or constrain retrievals. For retrievals of aerosol optical thickness (AOT) the differences between the ATLID L2a products (A-AER, A-EBD and A-ALD) and ACM-CAP are largely due to different treatments of measurement noise and the selection of volumes classified as aerosols. With its high degree of spatial smoothing, A-AER is biased low, while A-EBD and A-LAY are biased slightly high. The passive retrieval M-AOT performs very well in terms of random error, with a high degree of caution being taken to screening out cloudy profiles. In some areas MSI radiances may help to constrain ACM-CAP's aerosol retrievals, while the synergy with CPR may help to resolve ambiguities in the ATLID target classification, especially around weak backscatter targets and as the lidar becomes fully attenuated.

We have also included passive retrievals of ice and liquid cloud (M-CLD) at the nadir pixel; in the test scenes the volume of data for comparison are reduced by the fraction of daylit pixels and, for aerosols, the relative rarity of cloud-free profiles. In terms of IWP and LWP, M-CLD performed as well as ACM-CAP in profiles containing single-layered cloud scenes.

The simulated test scenes have proved invaluable as a resource for development of the L1 and L2 processor algorithms, the evaluation of the L2 products, and testing the processor chain. We stress that the present evaluation has been carried out with three simulated EarthCARE granules: that is 3/8ths of an EarthCARE orbit, which cannot be treated as representative of the global distribution or frequency of occurrence of all cloud and aerosol regimes. Furthermore, we are limited to the representation of the numerical models used to generate the scenes, which may not capture the full range of microphysical properties in the scenes that are sampled. Algorithm development will continue both prior to EarthCARE's launch and during operations; informed by calibration and validation activities, some of the challenges presented in this evaluation will be improved with the benefit of physical insights and testing with in-flight data.

In addition to the direct evaluation of geophysical quantities presented here, the ACM-RT and ACMB-DF processors (Barker et al., 2023) provide a radiative assessment of the retrievals against the on-board top-of-atmosphere radiative fluxes from the broadband radiometer (BBR). The insights gained from direct intercomparison of the composite L2a (ACM-COM) and the synergistic L2b (ACM-CAP) retrievals will help to interpret their performance in terms of broadband radiative closure.



Ongoing intercomparison between ESA L2 retrievals products, and with the corresponding Japanese L2 products, will be further leveraged for internal cross-verification of algorithms with in-flight EarthCARE data.

*Data availability.* The EarthCARE Level-2 demonstration products from simulated scenes, including all products discussed in this paper, are available from https://doi.org/10.5281/zenodo.7117115 (van Zadelhoff et al., 2023a)

*Author contributions.* SM led the L2 intercomparison. RH (ACM-CAP), DD and GJZ (A-ICE, A-EBD & A-AER), UW (A-ALD), PK & BPT (C-CLD), ZQ & JC (ACM-COM), AH (M-COP), and ND (M-AOT) contributed to aspects of the intercomparison and paper pertaining to their respective data products.

*Competing interests.* At least one of the (co-)authors is a member of the editorial board of Atmospheric Measurement Techniques and a guest member for the editorial board for the special issue "EarthCARE Level 2 algorithms and data products". The peer-review process was guided by an independent editor, and the authors have no other competing interests to declare.

### Acknowledgements

We thank Tobias Wehr, Michael Eisinger and Anthony Illingworth for valuable discussions, and are especially grateful to
600 Tobias Wehr for his support for this work over many years as ESA's EarthCARE Mission Scientist.



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
