# Peer review of "An intercomparison of EarthCARE cloud, aerosol and precipitation retrieval products"

_EGUsphere, 2023_

## Referee Comment (RC2)

Review of *"An intercomparison of EarthCARE cloud, aerosol and precipitation retrieval products"*, by Mason and coauthors, egusphere-2023-1682.

The approach used in this study, to evaluate the forthcoming EarthCARE cloud products by testing them against GEMS model output, is interesting and novel. Three test scenes from the model output are used, not unlike the Qu et al. (2022, amt-2022-300) article. As the authors note, the use of the model output provides the opportunity to perform a detailed evaluation of EarthCARE's instruments and retrieval algorithms when the model "truth" is known.

Microphysical properties-size distributions, ice and liquid water contents, etc, are generated in the model using Milbrandt and Yau's (2005) double-moment bulk cloud microphysics Scheme, which predicts mass and number mixing ratio for each of six hydrometeors classes: non-precipitating liquid droplets; ice crystals; rain; snow; graupel; and hail. These are the fundamental microphysical properties used in the evaluation of the EarthCARE instrument algorithms.

1. How good are the model variables and how well do they represent ice clouds? The retrievals strongly depend on the model data, and the fundamental question to ask is how good is the model data? There are many good field program data sets available, and have the model data become compared to recently collected microphysical observations from field programs? This obviously could be readily done. I looked back at the Milbrandt and Yau article and find the ice and snow categories and assumptions are based on the Ferrier and Ivanova et al. articles. The latter article uses FSSP particle probe data for the (bimodal) size distribution relationship(s). The FSSP data has been shown to significantly overestimate the concentrations of ice crystals because of instrument shattering. This could obviously affect the ATLID calculations, etc.

You note in Section 2 that apparent errors or biases in the retrievals presented in this paper may therefore be due to differences in assumptions underlying the model truth. In the concluding

remarks you note that your evaluation has been carried out with three simulated EarthCARE granules. I suggest that you discuss my point about model uncertainties in detail below that sentence. You could potentially do sensitivity studies with the model output, increasing the concentration of small ice crystals by an order of magnitude, etc, to see how the retrieval products would be affected.

2. Some of the acronyms used to represent the different instrument combinations and retrievals are not intuitive (ACM-CAP, etc) and are difficult to follow. I made my own table representing the acronyms. I suggest making a table containing the acronyms

3. Lines 159-163. Non-Rayleigh effects at W-band are extremely significant at reflectivities of 12 dBZ or so and above. W-band radars do not measure reflectivities above about 18 dBZ-that is, increasing "real" reflectivity results in decreasing W-band reflectivity. This should be mentioned. I looked at the Mroz (2023) article and it did seem like non-Rayleigh effects were accounted for.

4. General comment. Figure 1 and subsequent figures and in the text. Use g/m3, not kg/m3. The former is what is used in the literature and the units are such that values are easier to "digest". Lines 211, 233. Do you mean 1 $g/m^3$? 1 $kg/m^3$ is physically implausible.

5. Figure 2 is very informative and useful.

6. Line 274 and elsewhere. When referring to ice, use ice water content, when liquid, use liquid water content.

7. Lines 291-292. This statement is not quite correct. For W band, complete attenuation of the radar beam can occur in regions of very high radar reflectivity.

8. Section 3.2 What is not mentioned in your article is that a lidar beam is fully attenuated at an optical depth of about 3. Thus, in liquid cloud, penetration into the cloud layer would be a very short vertical distance. The relationship between optical depth and liquid water content (path) can be found in

   https://atmos.uw.edu/~robwood/papers/chilean_plume/optical_depth_relations.pdf.

This optical depth limitation applies to ice cloud as well. I suggest you mention this point in the text.

9. Looking at the Qu et al. (2022) article, what would the results have been if you used the Hawaii rather than the Halifax test scene? The precipitation rates and cloud ice water contents and optical depths would be considerably higher.

10. Figure 15 is extremely informative. Maybe just below the top row describing the retrieval, put in which instruments are being used.

---

## Author Response (AR1)

**Responses to review #1**

We thank the reviewer for their feedback, especially that which helped to restructure the paper and ensure the figures and errors were expressed clearly. We have implemented all suggested changes.

**Section 2.1: I think there should be more description of what the Halifax scene is here. I would really recommend adding a figure that shows the various truth geophysical fields here to set the context for the analysis that follows.**

We have added a paragraph to Section 2.1 describing all of the scenes in more detail, including highlighting that the Halifax scene covers the widest range of cloud, precipitation, and aerosol regimes. This shifts some detail from the introduction of the Results section. We also direct the reader to Qu et al. (2023) for more detail.

The various geophysical fields referred to as the model truth are provided as the first panel in all of the case study plots: we now re-iterate this in the introduction to the Results section.

**Figure 1: There is a significant volume of ice cloud in panel A with very low IWC (generally shaded in gray), for which there is no IWC produced by any product. Are these all areas that are below the detection sensitivity of the combined radar/lidar system? If so I would recommend mentioning this in the supporting text.**

This is indeed the case; we now reinforce this observation when discussing Fig. 1, and also Fig. 2, where a certain fraction of low-IWC features will remain unresolved by all products.

**Figure 2 and throughout the manuscript text: Maybe I'm being dense, but I don't understand the notation for RMS uncertainties, which have the form RMS error = +xx/-yy%. What does this mean? I would expect the RMS error to be a single number with units of kg/m^2.**

The relative errors should rightly have been referred to as the root mean squared logarithmic error (RMSLE) rather than RMSE, which in turn is expressed as a range of percentages for ease of interpretation. We now refer everywhere consistently to the "RMSLE", and have added a section to the start of Section 3 in which this formulation of the errors as a range of percentages is described more fully.

**Figure 3: Why does figure 3 not have a panel for C-CLD as in figure 2? Isn't ACM-COM derived from C-CLD and A-ICE?**

This is because C-CLD does not report ice effective radius, but rather snow characteristic diameter; the derivation of ice effective radius from the C-CLD retrieval is done in ACM-COM, as described in Cole et al. (2023). This is mentioned in the discussion, but we have now added a note in the caption to the figure as well.

**Having a section 3.1.1. 'Evaluation of retrieval uncertainties' seems out of place. This section evaluates uncertainties in IWC retrievals, which is fine. The thing I don't get is that the previous section entitled 'Ice clouds and snow' is written in nearly the same way; that is as an evaluation of retrieval uncertainty focused on other ice related variables such as IWP. Furthermore, the sections that (e.g. Liquid clouds) don't have a similar subsection. I suggest that you remove this sub section and just place this bit in the Ice clouds and snow section. In fact I would move much of this material to much earlier in that section to start with the IWC evaluation although I leave that up to you. Specifically I think Figure six should follow after Figure 1.**

Indeed, this section was a late addition and did interrupt the flow of the results section. We've made this change, shifting Figure 6 and the accompanying subsection to follow the IWP evaluation. Thanks for the suggestion.

**The first two paragraphs of 3.1.1 feel like they belong in the beginning of the ice clouds and snow section several pages earlier.**

We've made this change alongside the above, shifting these paragraphs to the introduction of Section 3.1

**Responses to review #2**

We thank the reviewer for their thoughtful comments, which have helped to improve the structure and clarity of the paper at key points. We've implemented all changes.

1. **How good are the model variables and how well do they represent ice clouds? The retrievals strongly depend on the model data, and the fundamental question to ask is how good is the model data? There are many good field program data sets available, and have the model data become compared to recently collected microphysical observations from field programs? This obviously could be readily done. I looked back at the Milbrandt and Yau article and find the ice and snow categories and assumptions are based on the Ferrier and Ivanova et al. articles. The latter article uses FSSP particle probe data for the (bimodal) size distribution relationship(s). The FSSP data has been shown**

**to significantly overestimate the concentrations of ice crystals because of instrument shattering. This could obviously affect the ATLID calculations, etc.**

**You note in Section 2 that apparent errors or biases in the retrievals presented in this paper may therefore be due to differences in assumptions underlying the model truth. In the concluding remarks you note that your evaluation has been carried out with three simulated EarthCARE granules. I suggest that you discuss my point about model uncertainties in detail below that sentence. You could potentially do sensitivity studies with the model output, increasing the concentration of small ice crystals by an order of magnitude, etc, to see how the retrieval products would be affected.**

These issues have been the topic of long discussions among the algorithm developers while the test scenes were developed and implemented, a process which itself informed the generation of the test scenes (Qu et al. 2023, https://doi.org/10.5194/amt-2022-300) and the simulation of the EarthCARE observations (Donovan et al. 2023, https://doi.org/10.5194/egusphere-2023-384). In several iterations, the scenes were iterated toward more realistic microphysics rather than having algorithm developers to tune toward the model physics. For example, Section 7 of Qu et al. (2023) describes how the ice from the Milbrandt and Yau scheme had to be modified to produce realistic ice number concentrations in order to facilitate realistic simulated EarthCARE measurements, just as you mention. Given the complexity of the generation of the test scenes, it is not now possible to run experiments that vary the scene microphysics within the scope of the present paper--although indeed some of the iterations on the test scenes over the years may have resembled just such an experiment. The algorithm description papers cite field campaigns and studies that are used to justify a-priori assumptions about cloud and precipitation physics, and further evaluation will take place using in-flight EarthCARE data as part of validation activities; we feel that these are the appropriate places for that analysis, rather than in the present paper.

Nevertheless, we can indeed improve the discussion of these issues in the conclusion:
- We now reiterate a point discussed in many of the algorithm description papers: that we do not wish to tune our retrieval assumptions toward the physics of the model.
- We now discuss in greater detail the model uncertainties in the conclusion; however, again, we are not able to include a sensitivity study based on the generated tests scenes within the scope of this paper.

2. **Some of the acronyms used to represent the different instrument combinations and retrievals are not intuitive (ACM-CAP, etc) and are difficult to follow. I made my own table representing the acronyms. I suggest making a table containing the acronyms**

Thank you for this suggestion, we've added Table 1 in which the product acronyms are expanded, following Tables 1 and 3 from Eisinger et al (2023).

3. **Lines 159-163. Non-Rayleigh effects at W-band are extremely significant at reflectivities of 12 dBZ or so and above. W-band radars do not measure**

**reflectivies above about 18 dBZ-that is, increasing "real" reflectivity results in decreasing W-band reflectivity. This should be mentioned. I looked at the Mroz (2023) article and it did seem like non-Rayleigh effects were accounted for.**

Indeed, non-Rayleigh effects become significant at W-band reflectivities of around 12 dBZ and above. The study by Mroz et al. (2023) takes these effects into account, as shown in their Figures A1 (a) and A2 (a). In W-band radar measurements, the reflectivity initially increases with the characteristic (melted equivalent) size of particles until it reaches a maximum value. After that point, further increases in particle size result in decreasing W-band reflectivity. This phenomenon occurs because larger particles have smaller backscattering cross-sections as opposed to $\sim D^6$ Rayleigh behaviour.

The lines in the figures represent a constant mass content of 1 g per $m^3$, which can lead to reflectivity values exceeding the 18 dBZ threshold. However, it's important to note that such high water content levels are quite rare in nature, and when they do occur, they are typically associated with strong attenuation, which can suppress the observed reflectivity values.

The line will now read "…the radar measurements are dominated by larger precipitating particles when present**, the microwave scattering properties of which are accounted for within the retrieval algorithms (Mroz et al 2023, Mason et al. 2023)."**

4. **General comment. Figure 1 and subsequent figures and in the text. Use g/m3, not kg/m3. The former is what is used in the literature and the units are such that values are easier to "digest". Lines 211, 233. Do you mean 1 g/m3? 1 kg/m3 is physically implausible.**

Indeed, this was a typo and should have read 1g/m3. As suggested, we have changed the figures and text throughout to use g instead of kg.

5. **Figure 2 is very informative and useful.**

Thank you.

6. **Line 274 and elsewhere. When referring to ice, use ice water content, when liquid, use liquid water content.**

Done

7. **Lines 291-292. This statement is not quite correct. For W band, complete attenuation of the radar beam can occur in regions of very high radar reflectivity.**

You're correct that in W-band radar, attenuation of the radar beam can occur in regions of very high radar reflectivity. To clarify, based on research by Protat et al. (2019), we would need to generate approximately 60 dB of attenuation to lose the radar signal from the surface. This calculation considers a surface return of 30-40 dBZ minus a sensitivity threshold of -35 dBZ.

To achieve this level of attenuation, one would need a 10 km layer of snow, corresponding to radar reflectivity values of around 20 dBZ. These conditions are indeed highly unlikely in practice. So while it's true that very high radar reflectivity can lead to attenuation, it's important to note that such extreme conditions are rare.

The line now reads, "… penetrates through **most** profiles of snow…"

8. **Section 3.2 What is not mentioned in your article is that a lidar beam is fully attenuated at an optical depth of about 3. Thus, in liquid cloud, penetration into the cloud layer would be a very short vertical distance. The relationship between optical depth and liquid water content (path) can be found in https://atmos.uw.edu/~robwood/papers/chilean_plume/optical_depth_relations.pdf. This optical depth limitation applies to ice cloud as well. I suggest you mention this point in the text.**

This is indeed worth stating explicitly. The introduction to section 3.2 now begins:
"**The penetration of the lidar beam into optically thick cloud layers is limited to around three optical depths.** The extinction of ATLID occurs over such a shallow layer in liquid clouds that no single-instrument ATLID retrieval of their properties is attempted…"

Indeed the A-ICE retrieval is based on just such a relation between the extinction and ice water content, but we can be more explicit about the coverage of the different products. In the first paragraph of Section 3.1 we now say:
"Single-instrument retrievals in the optical spectrum (i.e. ATLID and MSI) are sensitive to smaller ice clouds particles: A-ICE reports profiles of ice water content (IWC), extinction and effective radius **within the part of the cloud for which ATLID is not yet extinguished (around three optical depths)**, while M-COP reports ice water path (IWP), optical thickness and cloud-top effective radius **for the entire cloud layer**…"

9. **Looking at the Qu et al. (2022) article, what would the results have been if you used the Hawaii rather than the Halifax test scene? The precipitation rates and cloud ice water contents and optical depths would be considerably higher.**

The statistical figures include the Hawaii, Halifax and Baja scenes, so while the Halifax scene is used to illustrate the retrievals over the greatest range of regimes, the high values of precipitation rate, ice water contents and optical depths from the Hawaii scene are still represented within the distributions and joint histograms.

We have added to the second paragraph of the concluding discussion to reiterate the range of regimes across the three scenes that are covered in the statistical evaluation.

10. **Figure 15 is extremely informative. Maybe just below the top row describing the retrieval, put in which instruments are being used.**

Thank you. The vertical columns of this figure are labelled "MSI", "ATLID" and "CPR" to denote the instruments used. The L2a products are under one of these columns, while the synergistic products are displayed bridging these columns according to the instruments that help to constrain that aspect of the retrieval (i.e. MSI + ATLID + CPR for ACM-CAP ice and snow retrievals, but only MSI + ATLID for ACM-CAP's aerosol retrievals).